# Fat-Soluble Vitamins A, D, E, and K: Review of the Literature and Points of Interest for the Clinician

**DOI:** 10.3390/jcm13133641

**Published:** 2024-06-21

**Authors:** Emmanuel Andrès, Noel Lorenzo-Villalba, Jean-Edouard Terrade, Manuel Méndez-Bailon

**Affiliations:** 1Département de Médecine Interne, Hôpital de Hautepierre, Hôpitaux Universitaires de Strasbourg, Université de Strasbourg, 67000 Strasbourg, Francejean-edouard.terrade@chru-strasbourg.fr (J.-E.T.); 2Servicio de Medicina Interna, Facultad de Medicina, Instituto de Investigación Sanitaria del Hospital San Carlos, Hospital Clínico San Carlos, Universidad Complutense de Madrid, 28040 Madrid, Spain; manuelmenba@hotmail.com

**Keywords:** fat-soluble vitamins, vitamin A, vitamin D, vitamin E, vitamin K, metabolism, function, deficiency, etiology, diagnosis, treatment

## Abstract

Fat-soluble vitamins, including vitamins A, D, E, and K, are energy-free molecules that are essential to the body’s functioning and life. Their intake is almost exclusively exogenous, i.e., dietary. As a result, fat-soluble vitamin deficiencies are rarer in industrialized countries than in countries with limited resources. Certain groups of people are particularly affected, such as newborns or growing children, pregnant or breastfeeding women, and elderly or isolated individuals. Deficiencies in vitamins A, D, E, and K are also relatively frequent in subjects with digestive tract disorders, liver diseases, chronic pathologies, or in intensive care patients. Deficiencies or excesses of fat-soluble vitamins are responsible for a variety of more or less specific clinical pictures. Certain syndromes are typical of fat-soluble vitamin deficiency, such as the combination of ophthalmological and immunity impairments in the case of vitamin A deficiency or hemorrhagic syndrome and osteopenia in the case of vitamin E deficiency. This is also the case for osteomalacia, muscular weakness, even falls, and rickets in the case of vitamin D deficiency. Diagnosis of a deficiency in one of the fat-soluble vitamins relies on blood tests, which are not always essential for routine use. In this context, a therapeutic test may be proposed. Treatment of deficiencies requires vitamin supplementation, a well-balanced diet, and treatment of the cause.

## 1. Introduction

Vitamins are non-energy-rich molecules that are essential to the functioning of the body and to life itself. Depending on their physicochemical and metabolic properties, a distinction is made between water- and fat-soluble vitamins, the latter including vitamins A, D, E, and K [1]. Their intake is almost exclusively exogenous. As a result, vitamin deficiencies are rarer in industrialized countries than in those with limited resources.

Fat-soluble vitamin deficiencies are responsible for a variety of clinical pictures with varying degrees of specificity [1,2,3]. The ubiquitous and vital nature of the functions performed by vitamins, whether water- or fat-soluble, explains the wide variety of clinical manifestations and their potential severity [1,2].

Deficiencies in vitamins A, D, E, and K are found in individuals with special needs, such as growing children, pregnant or breastfeeding women, and the elderly [1,2,3,4]. They are also found in certain disease groups, such as digestive system diseases, chronic pathologies, and intensive care patients.

In practice, fat-soluble vitamin deficiencies are often underestimated due to a lack of understanding of the main clinical pictures or predominant clinical manifestations of the underlying etiologies of the deficiency.

The aim of this article is to highlight data of interest to the practitioner concerning fat-soluble vitamins, with particular reference to the most recent literature.

## 2. Materials and Methods

For this narrative review, we searched the medical literature for complete published studies on the management of patients with fat-soluble vitamin deficiencies. A literature search of the *PubMed* (through Medline) electronic database was carried out with a time limit of 5 years. Only studies published in English and French were considered. The medical subject heading (MeSH) and keywords used were “fat-soluble vitamin deficiency”, “vitamin A deficiency”, “vitamin D deficiency”, “vitamin K deficiency”, “clinical manifestations”, “etiology”, “management”, and “treatment”. The literature analysis included only articles on human subjects.

We also screened the reference lists of the most relevant articles to identify other studies not captured in the initial literature search with *Google Scholar*.

A number of reference articles on this topic were also used and cited in the references, even though they date from the last twenty years.

Textbooks of internal medicine, general medicine specialty, and nutrition, as well as the reports of academic learned societies and governmental organizations were also used in this work.

## 3. Metabolism of Fat-Soluble Vitamins

There are 13 vitamins [1]. They can be divided into two groups according to their physicochemical and metabolic properties and the nature of their physiological functions [1,2]. The fat-soluble vitamin group comprises four vitamins: A (retinol), D (calciferol), E (tocopherol), and K (phylloquinone [K1] and menaquinone [K2]). The nine water-soluble vitamins include B1 (thiamine), B2 (riboflavin), B3 (niacin), B5 (pantothenic acid), B6 (pyridoxine), B8 (biotin), B9 (folic acid or folate), B12 (cobalamin), and C (ascorbic acid).

For fat-soluble vitamins, in particular vitamins A, E, and K, intake is exclusively exogenous, i.e., dietary [1,2,3]. For vitamin D, intake is both endogenous and exogenous [1]. In the case of vitamin D, exposure to ultraviolet B (UVB) rays converts 7-dehydrocholesterol located in the skin into provitamin and then into vitamin D3 [1,3]. Hepatic hydroxylation at position 25, followed by renal hydroxylation at position 1, produces 1,25-dihydroxy-vitamin D3, the active form of vitamin D.

Recommended intakes of fat-soluble vitamins A, D, E, and K vary according to age, gender, and physiological situation, such as pregnancy or breastfeeding [1,4]. For adults, the recommended intake of vitamin A is generally between 700 and 900 micrograms (µg) retinol equivalents per day for men and between 600 and 700 µg retinol equivalents per day for women [1,5]. For adults, vitamin D intake is generally estimated at between 600 and 800 international units (IU) per day [1,6]. The recommended intake of vitamin E for adults is generally between 15 and 30 milligrams (mg) per day [1,7]. Vitamin K intake is generally between 90 and 120 µg per day for men and between 70 and 90 µg per day for women [1,8].

It should be stressed that vitamin D is different from other fat-soluble vitamins, so much so that it is considered a pro-hormone because it has various active and inactive metabolites and because its activation responds to feedback from blood calcium levels [1].

Fat-soluble vitamins A, D, E, and K are fat-dissolving vitamins that are absorbed by the small intestine along with dietary fat [1,4]. Absorption of fat-soluble vitamins is predominantly proximal in the small intestine, except for vitamin K2, whose absorption predominantly takes place in the colon [5,8]. Absorption occurs via an active mechanism for vitamins A and K and by the facilitated diffusion for vitamins E and D. In all cases, as with lipids in general, good enterocyte, hepatobiliary, and pancreatic functions are essential to ensure the correct absorption and bioavailability of fat-soluble vitamins [1].

Vitamin A is present in food in the form of retinol or provitamin carotenoids [1,9]. In the small intestine, retinol and carotenoids are absorbed by intestinal cells and converted into retinol. Retinol is then bound to a transport protein, called the retinol carrier protein (RBP), and transported to the liver for storage or to other tissues for utilization.

Vitamin D3 (the active form of vitamin D) is produced in the skin by the sun’s UVB rays or absorbed from food or supplements [1,6]. In the small intestine, vitamin D is absorbed with dietary fat. It is then converted to an active form, 1,25-dihydroxy-vitamin D3, in the liver and kidneys before being used in the body to regulate calcium and phosphorus metabolism.

Vitamin E is present in food in the form of tocopherols and tocotrienols [1,7]. In the small intestine, tocopherols and tocotrienols are absorbed by intestinal cells using specific transporters. Once absorbed, vitamin E is transported in the blood by lipoproteins and distributed to the tissues for utilization.

Vitamin K1 is found in green leafy vegetables, while vitamin K2 is found in animal products and fermented foods [1,8]. In the small intestine, vitamin K is absorbed along with dietary fat. Vitamin K is then transported in the blood by lipoproteins and distributed to tissues for utilization, particularly in the liver, for the synthesis of blood clotting factors.

The elimination of fat-soluble vitamins A, D, E, and K is mainly fecal, with the excess being eliminated in the stool via the bile, but also urinary for a proportion of vitamins A and E, as well as a small vitamin D proportion [1,9]. As part of homeostasis maintenance, all vitamins are stored in the liver in anticipation of later use and to avoid an excess of these vitamins, which could be toxic for the organism [1,4].

## 4. Functions of Fat-Soluble Vitamins

The physiological functions of fat-soluble vitamins in the body are numerous [5,6,7,8,9]. These functions explain most of the clinical manifestations caused by fat-soluble vitamin deficiency.

Vitamin A is essential for vision, especially night vision [5], and is required for the synthesis of rhodopsin, a visual pigment present in retinal cells. It is very important for normal growth and development, including cell differentiation and bone growth regulation [4,5]. Vitamin A also plays a role in immune function, maintaining mucosal integrity and supporting immune cell differentiation and function [5]. Retinol also impacts growth. Finally, retinoic acid plays an important role in reproduction, more specifically in spermatogenesis, gestation, and fetal development.

Vitamin D is essential for bone mineralization, promoting intestinal calcium and phosphorus absorption, and regulating calcium metabolism in bone [4,6]. Comparable to vitamin A, it regulates immune function, including the modulation of the inflammatory response and stimulation of antimicrobial peptide production [10,11]. Vitamin D is also involved in the regulation of various metabolic processes, including blood sugar regulation and the modulation of muscle and heart cell function [5]. Moreover, a direct role of this vitamin in genomic processes has been documented in recent years [12].

Vitamin E is a powerful antioxidant that protects cell membranes and lipids against oxidative damage caused by free radicals [1,7]. Vitamin E can help reduce the risk of cardiovascular disease by reducing oxidation of low-density lipoprotein (LDL) cholesterol and improving endothelial function (anti-atheromatous action) [7]. Vitamin E may have beneficial effects on the skin’s integrity by reducing inflammation, improving wound healing, and protecting against sun damage and free radicals [7]. It may also have a protective role against cancer development.

Vitamin K is essential for the synthesis of blood clotting factors, which is necessary for normal blood coagulation and the prevention of excessive bleeding [4,8]. In particular, vitamin K is required to activate certain coagulation factors, notably factors II (prothrombin), VII, IX, and X. It is also involved in bone mineralization, regulating the function of proteins involved in calcium binding to bone [8]. Finally, vitamin K may play a role in regulating lipid and glucose metabolism, although the precise mechanisms are not yet fully understood [10,11].

The daily requirements, safety limits, main dietary sources, and physiological roles of fat-soluble vitamins are summarized in Table 1 [1,5,6,7,8].

## 5. Epidemiological Data on Fat-Soluble Vitamin Deficiencies

Epidemiological data on the deficiencies of fat-soluble vitamins A, D, E, and K vary depending on the region of the world, the population studied, and the data collection methods used [1,4]. Relatively few solid data are available from industrialized countries due to a lack of consistent studies on this topic, particularly for vitamins E and K. Most studies available in the current literature focus on specific populations or patient groups.

However, it should be emphasized that vitamin A deficiency remains a major public health issue in many developing regions, particularly in sub-Saharan Africa and South Asia [5,13]. According to the World Health Organization, around 250 million children worldwide are vitamin A deficient, leaving them vulnerable to infections, night blindness, and other serious complications.

The prevalence of vitamin D deficiency varies between populations but is common in many parts of the world, including developed countries, particularly in areas with limited sun exposure [6,14]. In the latter, groups at high risk of vitamin D deficiency include the elderly and/or institutionalized, dark-skinned people, obese patients, people living in high latitudes or with limited sun exposure, as well as people with certain chronic diseases, such as chronic renal failure.

## 6. Clinical Manifestations of Fat-Soluble Vitamin Deficiencies

The clinical presentation and diagnostic approach differ according to whether the condition is global malnutrition or undernutrition, or rather a selective vitamin deficiency, in this case, of vitamins A, D, E, and K [1,4].

In industrialized countries, vitamin, and nutritional deficiencies in general tend to be selective (Table 2) and more related to atypical dietary habits or underlying pathologies [1,5]. In this context, the clinical manifestations point towards a type of water- and fat-soluble vitamin deficiency (Table 2) [5,6,7,8].

### 6.1. Vitamin A Deficiency

A vitamin deficiency generally refers to a relative lack of a vitamin in the body [1]. It can be accompanied by frustrating clinical manifestations, giving rise to the concept of “subtle” fat-soluble deficiency. The scientific and medical communities do not universally recognize the latter. Vitamin deficiency occurs when levels of a vitamin in the body are so low that they cause symptoms, clinical signs, and adverse health effects.

A schematic distinction is made between, on the one hand, vitamin deficiencies resulting in characteristic syndromic associations that immediately suggest the diagnosis, as illustrated in Table 2, and, on the other hand, deficiencies whose polymorphism and widespread clinical manifestations make it difficult to provide a targeted diagnostic orientation [1,5,6,7,8].

Vitamin A plays a vital role in maintaining epithelial integrity and cell differentiation [1,5]. In this context, the combination of ophthalmopathy and immune deficiency is characteristic of vitamin A deficiency and reflects the impact of the impairment of these two functions at ocular and blood levels (Table 2). In practice, vitamin A is essential for normal vision [4,5]. Vitamin A deficiency leads to a more or less marked reduction in night vision (night blindness), which translates into an inability to see in low-light conditions and is accompanied by photophobia. Reduced visual acuity can progressively lead to permanent night blindness or total blindness. The physical signs are essentially xerophthalmia, linked to reduced tear production, and are indistinguishable from Sjögren’s syndrome and keratomalacia [1,5]. Ophthalmological examination reveals characteristic conjunctival lesions of triangular perilimbal debris [5]. A fluorescein test is required to detect punctiform ulcerative keratopathy, more commonly associated with the clinical picture of ocular herpes or even ophthalmic herpes zoster.

Predisposition or susceptibility to infection is the other main clinical manifestation of vitamin A deficiency [5,10,11]. It results from a combination of defective medullary leukocyte differentiation and damage to the epithelial mechanical barrier, particularly in the skin.

In cases of vitamin A deficiency, skin and digestive mucosa disorders are frequent, characterized by dry skin, hyperkeratotic papular lesions symmetrically located on the lateral and dorsal surfaces of the limbs, near the knees and elbows, and diarrhea without specificity [1,4,5]. Vitamin A deficiency can also cause itching and skin ulcers.

Epidemiological studies have also shown a high anemia prevalence in individuals with vitamin A deficiency [10,11]. The mechanism remains unknown and is probably multifactorial. It is important to note that vitamin A deficiency is generally not the sole cause of anemia but can aggravate an existing anemia or contribute to its development. Other factors, such as iron, vitamin B12, or folic acid deficiency, as well as chronic diseases or genetic disorders, may also play a role in anemia development [15].

Vitamin A deficiency can also affect the growth and development of children and infants, leading to stunted physical growth and delayed mental development [5]. In this context, it is typically associated with recurrent bronchial infections.

### 6.2. Vitamin D Deficiency

Osteomalacia and fractures are schematically the two main clinical manifestations of symptomatic vitamin D deficiency in adults (Table 2) [6,16]. Osteomalacia causes diffuse and mechanically induced bone pain, predominantly in the pelvis, thorax, and spine. X-ray findings include increased bone transparency, biconcave vertebral compression, and Looser–Milkmann striae, which correspond to small fissures perpendicular to the bone cortex, preferentially affecting the pelvis and femur [6]. In children, it manifests as rickets [1].

Vitamin D-deficient patients also often present with fatigue or asthenia, muscular fatigue and weakness, reduced muscle strength, and a predisposition to falls [6,10,11]. A waddling gait is usually observed, associated with muscular weakness and reduced muscle coordination. In elderly subjects, treatment with vitamin D (at least 800 IU/day) combined with calcium has been shown to reduce the relative risk of falls and the consequences of bed rest.

The risk of developing several pathologies in the presence of vitamin D deficiency has also been mentioned in some studies but is currently far from documented (based essentially on epidemiological studies):Cancer, especially colorectal and breast cancer, but also prostate and lung cancer;Myelodysplastic syndromes;Arterial hypertension, with an increased risk of cardiovascular events;Autoimmune pathologies such as multiple sclerosis, rheumatoid arthritis, systemic lupus erythematosus, and Type 1 diabetes in children;Osteoarthritis;Schizophrenia and depression [10,11].

The exact mechanisms through which vitamin D might influence the development and progression of these various pathologies are, to date, not fully known or understood. However, it is important to note that the association between vitamin D deficiency and, for example, cancer and/or autoimmune diseases does not necessarily imply a causal relationship. Indeed, most interventional studies on the subject have been inconclusive (lack of clinical benefit in terms of prevention).

Finally, recent studies suggested a potential role for vitamin D deficiency in chronic asthenia, anxiety, depression, and even fibromyalgia [10,11]. It is classically accepted that vitamin D deficiency can lead to muscular weakness and generalized fatigue. In this context, it has been suggested that vitamin D may contribute to chronic asthenia and fatigue in fibromyalgia. Studies have also reported that vitamin D may have anti-inflammatory and analgesic effects, which could be relevant to patients with fibromyalgia. It should be noted, however, that these disorders are complex and multifactorial conditions, with many potential contributing factors, including genetic, environmental, and psychosocial factors.

### 6.3. Vitamin E Deficiency

Neurological and ocular signs dominate the clinical picture of vitamin E deficiency (Table 2) [1,7]. Symptomatic deficiency onset is usually characterized by diminished reflexes, impaired proprioceptive and vibratory sensitivity, reduced muscle strength in the distal region, and sometimes night blindness. Ataxia and nystagmus may also be present. At a later stage, ocular signs progress to ophthalmoplegia and blindness, with the onset of cognitive disorders indicating the depth and age of the deficit. The earlier the diagnosis and treatment, the greater the chance of recovery [7].

Weakening of the immune system, with increased susceptibility to infection, has been reported, as well as cases of hemolytic anemia, especially in the case of hemolytic pathologies such as beta-thalassemia major, sickle cell anemia, and hereditary spherocytosis [1,7,10]. Finally, vitamin E deficiency has been linked to fertility problems in both men and women [7,10,11].

### 6.4. Vitamin K Deficiency

Spontaneous hemorrhagic syndrome or hemorrhagic syndrome following minor trauma dominates the clinical picture of vitamin K deficiency (Table 2) [1,5,8]. All body sites may be affected. Mucocutaneous hemorrhages may be observed (e.g., epistaxis, gingivorrhagia, or purpura), as well as menometrorrhagia, hematuria, muscle ecchymosis and hematomas, gastrointestinal bleeding, or even more serious cerebral or deep visceral hemorrhages. In this context, an association with vitamin C deficiency (scurvy) should not be overlooked [17].

Importantly, vitamin K is also essential for the synthesis of certain bone matrix proteins, including osteocalcin, and for the balance between osteoblasts and osteoclasts [1,8]. Vitamin K deficiency can disrupt this balance, contributing to bone mineral density reduction, osteopenia, and increased fracture risk. The combination of osteopenia and bleeding syndrome is highly suggestive of vitamin K deficiency [8]. Epidemiological studies have suggested an association between low vitamin K intake and an increased osteoporosis risk [1,8].

A role for vitamin K in muscle function has also been mentioned by authors, explaining some of the muscle weakness observed in vitamin K deficiency [8,10,11].

Rare cases of cutaneous necrosis have been described upon initiation of anti-vitamin K (AVK) anticoagulant therapy, notably in patients with protein C deficiency [1,8].

### 6.5. Multiple Fat-Soluble Vitamin Deficiencies

In cases of overt malnutrition or undernutrition, vitamin deficiencies are generally multiple, with a non-specific clinical picture dominated by massive weight loss, leading to pseudo-neoplastic cachexia or even kwashiorkor in the most extreme cases [4,5]. In this context, these conditions and clinical pictures are mainly observed in developing countries.

In industrialized countries, global malnutrition or malnutrition may exceptionally be observed in certain pathologies, such as refractory Crohn’s or celiac disease. They may also be observed in elderly people, particularly those living in isolation, as well as in institutionalized patients, those suffering from chronic pathologies, and intensive care patients, especially those with multi-visceral failure and extensive burns [1,4,5]. In this context, it should be noted that the manifestations of the underlying disease often predominate over those of fat-soluble vitamin deficiencies, which can complicate the diagnosis of the latter and cause the deficiency to go unnoticed.

## 7. Etiologies of Fat-Soluble Vitamin Deficiencies

Deficiencies in fat-soluble vitamins A, D, E, and K may have a variety of causes and etiologies, with sometimes multiple causes for a specific deficiency [1,4]. In industrialized countries, the most common etiologies of fat-soluble vitamin deficiency are diet, malabsorption, and liver disease.

### 7.1. Dietary Deficiencies

A diet low in food sources rich in fat-soluble vitamins can lead to deficiencies. For example, low consumption of green leafy vegetables, beta-carotene-rich fruits (for vitamin A) [5], oily fish (for vitamin D) [6], nuts and seeds (for vitamin E) [7], or cruciferous vegetables (for vitamin K) [8] can contribute to fat-soluble vitamin deficiency (Table 1).

In industrialized countries, it should also be pointed out that people in situations of socio-economic insecurity and isolation often have limited access to high nutritional quality foods [1,18]. This fact is particularly observed in institutionalized elderly patients, isolated subjects such as homeless individuals, and single-parent families. They are in a state of food insecurity, with more or less selective vitamin deficiencies.

Today, several new phenomena are also responsible for multiple vitamin deficiencies, which have been observed for some years but have been exacerbated by the COVID-19 pandemic [18]. We are witnessing a “junk food” epidemic, with excessive intakes of sugars and fats responsible for an obesity epidemic, metabolic syndromes, and diabetes. Paradoxically, the latter is accompanied by deficiencies—sometimes selective, often multiple—in vitamins and trace elements linked to the quality of the food consumed.

This is also the case for individuals following “unconventional” dietary practices. In this context, they may develop more or less selective deficiencies in various vitamins. This is the case, for example, for individuals practicing veganism and vegetarianism, but also “fad diets” (ketogenic diet, paleo diet, intermittent fasting, gluten-free diet, and other strict exclusion diets).

### 7.2. Digestive Tract Disorders and Malabsorption

Gastrointestinal disorders impairing fat absorption can lead to malabsorption of fat-soluble vitamins [1,4]. Examples include celiac disease, Crohn’s disease, ulcerative colitis, Whipple’s disease, chronic pancreatitis (e.g., cystic fibrosis), and chronic cholestasis (e.g., primary biliary cirrhosis).

Several surgical procedures can also affect the absorption of fats and fat-soluble vitamins, such as bowel resections, gastrectomy (even partial), and bariatric surgery, regardless of the surgical setup (sleeve gastrectomy, gastric bypass, or biliopancreatic bypass) [5,6,7,8].

Gastrointestinal disorders can impair fat absorption through a variety of mechanisms, including a reduced surface area for absorption, with damage to intestinal villi (e.g., in celiac disease, Crohn’s disease) or intestinal resection (e.g., in short bowel syndrome) [1,4]. They can also affect fat absorption through decreased digestive enzyme production (e.g., in chronic pancreatitis, cystic fibrosis); bile salt disruption with damage to the ileum (e.g., in Crohn’s disease, short bowel syndrome); and intestinal dysbiosis (disturbance of intestinal flora) [1,4].

### 7.3. Liver Diseases

Liver diseases affecting liver function or fat storage disorders, such as cirrhosis, chronic hepatitis (chronic alcoholism, chronic viral hepatitis B and C, autoimmune hepatitis, hemochromatosis, and Wilson’s disease), metabolic dysfunction-associated steatohepatitis (MASH), and ex-nonalcoholic steatohepatitis ([NASH], also known as fatty liver) can disrupt the metabolism of fat-soluble vitamins [1,4].

Liver diseases disrupt fat metabolism by affecting bile production and secretion, lipoprotein metabolism, beta-oxidation of fatty acids, cholesterol and triglyceride syntheses, and ketone body production [1,4]. These disturbances can lead to fat accumulation in the liver, fat malabsorption, and systemic lipid imbalances, influencing the patient’s overall health. 

All liver diseases accompanied by chronic cholestasis, such as primary biliary cirrhosis and sclerosing cholangitis, may also be accompanied by a deficiency of fat-soluble vitamins [1,4].

### 7.4. Other Common Conditions and Disorders

Certain physiological or pathological conditions can lead to an increased use of fat-soluble vitamins A, D, E, and K, which can lead to deficiencies if needs are not met by diet or adequate supplementation. Pregnancy, breastfeeding, and rapid growth in children, for example, call for special attention and even supplementation if the diet is unbalanced or insufficiently rich in vitamins and other trace elements.

The need for fat-soluble vitamins is also increased in cases of hyperthyroidism, as well as chronic diseases such as chronic heart, renal and/or respiratory failure, cancer, and intensive care patients (patients with multi-visceral failure, burn victims, and exclusive parenteral nutrition) [1,5,6,7,8]. 

This is also the case for diseases causing various protein losses (e.g., nephrotic syndromes and exudative enteropathies) and chronic inflammatory syndromes (e.g., Sjögren’s syndrome, systemic lupus erythematosus, rheumatoid arthritis, acquired immunodeficiency syndrome, septic shock, etc.), which interfere with the synthesis of carrier proteins or require higher levels of fat-soluble vitamins.

Some medications may increase the need for fat-soluble vitamins and/or interfere with the absorption or metabolism of fat-soluble vitamins. These include orlistat and sibutramine, corticosteroids, cholestyramine and colestipol, antiepileptic drugs (such as phenytoin, phenobarbital, and carbamazepine), and mineral oils, particularly used as laxatives [5,6,7,8].

### 7.5. Etiologies of Specific Fat-Soluble Vitamin Deficiencies

Certain parasitic infections, such as *Trichuris trichiura* and *Ascaris lombricoides*, can affect the bioavailability of vitamin A in the body by disrupting its absorption and metabolism [5].

Malnutrition, particularly among children in developing regions, is a major cause of vitamin A deficiency [5]. Regions where the diet is low in vitamin A are particularly prone to high vitamin A deficiency rates. The latter is exacerbated by concomitant zinc deficiency.

Insufficient sun exposure due to factors such as living in areas with little sunshine, wearing covering clothing, using sunscreen, or being confined indoors can lead to vitamin D deficiency [6].

Certain rare genetic conditions affecting the specific metabolism of a vitamin can lead to a deficiency of that vitamin [7]. Ataxia with vitamin E deficiency (AVED) and alpha-tocopherol transferrin deficiency (a protein that transports vitamin E in the blood), for example, can lead to an inability to absorb or use vitamin E effectively.

Some antibiotics, particularly broad-spectrum antibiotics (e.g., third-generation cephalosporins), can disrupt the bacterial synthesis of vitamin K in the gut by altering the intestinal flora, which can lead to vitamin K deficiency [8].

Finally, newborns are often vitamin K-deficient at birth due to the limited passage of this vitamin through the placenta and the low vitamin K concentration in breast milk [1,8]. For this reason, they usually receive a vitamin K injection shortly after birth to prevent bleeding complications.

## 8. Diagnosis of Fat-Soluble Vitamin Deficiencies

To diagnose a vitamin deficiency, whether water- or fat-soluble, the practitioner will begin by questioning the patient about his or her medical and surgical history and dietary habits [1,4].

He/she will then carry out a complete physical examination, looking for the signs described above, such as the signs associated with ophthalmopathy and immune disorders, osteopenia, bleeding manifestations, etc. (Table 2). 

As part of the diagnostic approach, the practitioner must consider the clinical context, as well as the situations and sites at risk of fat-soluble deficiencies, to avoid having an overly long list of biological tests.

Vitamin deficiency confirmation, in this case in the vitamins A, D, E, and K, theoretically involves a series of direct and indirect assays performed on the blood (Table 3) [1,4]. For the assay to be interpretable, the sampling and storage/transport conditions must respected (storage protected from light and refrigerated). The determination of fat-soluble vitamin levels is generally performed using high-performance liquid chromatography (HPLC) or direct fluorimetric techniques.

In practice, it should be noted that these examinations are not always used and often give way to therapeutic tests, particularly in the case of a typical presentation with a strong clinical presumption [19].

This seems even more necessary, given the over-prescription of vitamin assays in routine clinical practice. For example, this is the case with vitamin D (D5-OH-D), which, according to several medical societies, should be reserved for very specific clinical situations: assessment and management of elderly people suffering from repeated falls, before and after bariatric surgery, treatment with a drug whose summary of product characteristics recommends vitamin D dosage (bis-phosphonates), follow-up of adult kidney transplant patients beyond 3 months post-transplant, rickets diagnosis, and osteomalacia diagnosis. However, some experts recommend a vitamin D dosage in all clinical situations where there is a risk of osteopathic fragility. In this context, biological phosphocalcic tests are essential, as well as medical imaging (e.g., bone X-rays or scans), to assess the bone density and detect abnormalities associated with vitamin D deficiency [6].

In practice, to avoid unnecessary vitamin D testing, it is essential to follow clear guidelines, target-test at-risk populations, prescribe tests thoughtfully, and educate patients about the appropriate indications [1,4]. Above all, it is essential to reassess the need for repeat testing in patients who have already been diagnosed and treated for vitamin D deficiency based on their response to treatment and the absence of persistent symptoms. Routine tests should also be avoided in healthy individuals.

The help of a dietician may be required for a detailed assessment of the patient’s diet and lifestyle habits [1,4]. In this context, it is important to note that the nutritional values of certain common foods given in the nutrition fact labels should be considered with caution. Values may vary according to factors such as preparation method, food variety, packaging, state of freshness, etc.

## 9. Treatment of Fat-Soluble Vitamin Deficiencies

Treatment of fat-soluble vitamin deficiencies depends on several factors, including the deficiency type, its severity, and the patient’s individual needs (growth, pregnancy, breastfeeding, and underlying diseases) [1,4,19].

Treatment is based on a few key intangible principles: vitamin supplementation, balanced diet, assessment and treatment of underlying causes, regular medical follow-up, education, and prevention. The most common method to treat a fat-soluble vitamin deficiency is supplementation with specific preparations of vitamins A, D, E, or K [1,5,6,7,8].

Vitamin deficiency treatment must be individualized according to the specific needs and circumstances of each patient [5,6,7,8]. This is even more true in the age of personalized medicine, which must also be participatory, preventive, predictive, and proven (5P medicine).

Implementing individualized protocols to treat vitamin deficiencies is complex but essential for optimal patient management [1,19]. These protocols must take into account the variability of individual needs, the complexity of interactions between nutrients, the need for an accurate diagnosis of vitamin deficiency, patient compliance, and, more generally, economic considerations and the variability of care systems [5,19]. Against this backdrop, the need for forward-looking clinical studies has never been greater. Continuing education, interdisciplinary collaboration, and the use of information and communication technologies are key to overcoming these challenges.

It should be emphasized that the treatment regimens proposed for this supplementation are mostly based on studies that do not meet the evidence-based medicine criteria [18]. Table 3 shows the tried-and-tested treatment regimens, often used in France and in Europe, for various fat-soluble vitamin deficiencies [1,19,20,21].

Vitamin A, D, E, and K supplements can be taken orally in the form of tablets, capsules, or drinkable preparations, but they can also be administered intramuscularly or intravenously, according to the frequency and therapeutic regimen, depending on the severity of the deficiency [5,6,7,8,19]. 

Wide varieties of curative therapeutic regimens have been proposed by medical academic societies (e.g., endocrine, rheumatologic, and internal medicine societies). Nevertheless, to date, the optimal therapeutic regimen for vitamin D supplementation is currently unclear [21]. However, much of the evidence in the literature suggests that, regardless of the dosing schedule (daily, weekly, or monthly), the same results are obtained, providing that all potential factors affecting the inter-individual variation are taken into consideration. In the recent literature, administration of a daily cholecalciferol dose of 1500–2000 IU per day has been recommended for the treatment of vitamin D deficiency, in particular, to reach serum vitamin D levels of >30 ng/mL [21]. However, with this therapeutic regime, the correction of severe hypovitaminosis D (serum levels < 10 ng/mL) or overt osteomalacia requires several months. In this setting, doses of 3000 to 10,000 daily IU cholecalciferol (on average 5000 IU/day) (or equivalent weekly) have been suggested over 1–2 months to normalize the serum level within a few weeks. Alternatively, Bertoldo et al. recommend a cholecalciferol load dosage of 100,000–150,000 IU, followed by a maintenance dose of 2000 IU/day [21]. For practice, it is interesting to note that several recent studies showed daily administration regimens to be more promising in terms of both skeletal (especially when associated with calcium supplementation) and extra-skeletal outcomes [21]. 

To prevent vitamin D deficiency, the recommended daily intake for adults is around 400 IU between 51 and 70 years old and 600 IU after the age of 70. This dose should be increased to between 800 and 1000 IU when sun exposure is insufficient [1,6].

A balanced and nutrient-rich diet is essential to help make up for the deficiencies in both water- and fat-soluble vitamins [1,19]. Vitamin-rich foods should be included in the daily diet, with an emphasis on fruits, vegetables, whole grains, dairy products, lean meats, fish, and plant-based protein sources. Patients with vitamin deficiencies should be followed up regularly by a healthcare professional to assess their response to treatment, possibly monitor blood vitamin levels, and adjust their supplement doses if necessary [1,4].

Appropriate education on the importance of a balanced diet and the prevention of vitamin deficiencies can help prevent future deficiencies. Patients should be informed about vitamin-rich foods and encouraged to adopt a healthy lifestyle [1,5,19]. They need to take ownership of their disorder, with therapeutic education techniques proven in other metabolic diseases if needed, such as diabetes.

A medical consultation is necessary to properly assess and monitor vitamin deficiency and recommend an appropriate treatment plan [20].

Hypervitaminosis can occur when the body accumulates excessive vitamin amounts, particularly fat-soluble vitamins [20,22]. This can be caused by their consumption over a long period of time of high doses of vitamin supplements, excessive levels of vitamin-enriched foods, or food supplements. For all vitamins, especially fat-soluble ones, it is essential not to forget the hidden sources, notably in nutritional supplements, “diet” and “wellness” products found on the Web, in drugstores, etc. 

The above toxic effects described are linked to the use of the active metabolite of vitamin D, namely calcitriol. This is usually not the case with the use of vitamin D3 supplementation, except in cases of other causes of associated hypercalcemia (e.g., use of thiazidic diuretics, use of lithium, Burnett’s syndrome). 

For example, the clinical manifestations of hypervitaminosis D are related to induced hypercalcemia [22]. They include asthenia, anorexia, weight loss, digestive disorders, and neurological signs, such as headaches with vomiting [20,22]. Arterial hypertension, polyuro-polydypsic syndrome, and renal failure have also been reported [4,6]. Urinary lithiasis may also occur in cases of chronic overload, possibly leading to nephrocalcinosis, as well as calcification of the vessels (particularly in the case of excess in chronic renal failure) and the heart [4,20].

Infants are more sensitive to excess vitamin D than adults. In children, excess vitamin D can lead to hypercalcemia and brain and/or kidney damage and can cause abnormal tooth structure. The symptomatology is sometimes more “spectacular”, with convulsions, for example [1]. Fetal malformations (aortic stenosis) have been reported in pregnant women [4,7].

To avoid hypervitaminosis, it is crucial to carefully assess the vitamin requirements, educate patients, prescribe appropriate doses, and regularly monitor patients’ health [20,22]. Healthcare professionals must adopt prudent prescribing practices, ensure clear communication between caregivers, and react promptly to signs of vitamin toxicity.

## 10. Conclusions

In conclusion, fat-soluble vitamins play a crucial role in the proper functioning of our bodies and in many aspects of human health, from vision to bone health and immunity. As these vitamins cannot be synthesized by the body, they must be obtained from dietary sources. A balanced diet, including a variety of foods rich in fat-soluble vitamins, is essential to ensure adequate intake. This is particularly the case during growth, in elderly or isolated people, and during pregnancy or breastfeeding. The absorption and metabolism of fat-soluble vitamins require particular attention, as any imbalance can have harmful consequences for health. Damage to the digestive system, liver diseases, and many other conditions, such as chronic illnesses, can lead to more or less severe and selective deficiencies in these vitamins. However, it is also important not to overlook the risks associated with vitamin over-consumption, as excess can lead to serious complications. Ultimately, a balanced approach, based on a varied diet and professional advice, is essential to maintain an optimal fat-soluble vitamin status and promote sustainable global health.

## Figures and Tables

**Table 1 jcm-13-03641-t001:** Recommended intakes, dietary sources, and physiological functions of fat-soluble vitamins [1,5,6,7,8].

Vitamins	Food Sources	Recommended Dietary Allowance	Toxic Dose	Physiological Functions
A	Fish oil, liver, butter, egg yolks, carrots, sweet potatoes, and spinach	600–950 RE/day (1 μg of retinol = 3.3 IU = 6 μg of beta-carotene)	3000 RE/day	Vision, immunity, and growth
D	Cod liver oil, egg yolks, fatty fish (salmon, mackerel, sardines), eggs, liver, and mushrooms	5–10 IU/day	1000 IU/day	Calcium homeostasis, immunity, and genomic regulation
E	Green vegetables, wheat germ, vegetable oils (wheat germ oil, sunflower oil), nuts and seeds (almonds, sunflower seeds), avocado, and spinach	12 mg/day	–	Antioxidant action
K	Green leafy vegetables (kale, spinach, broccoli), cruciferous vegetables (cauliflower, Brussels sprouts), liver, egg yolks, soybean oil, meat, and dairy products	0.1–1 μg/Kg/day	–	Hemostasis and bone metabolism

IU: international unit; RE: retinol equivalent.

**Table 2 jcm-13-03641-t002:** Main clinical manifestations of fat-soluble vitamin deficiency [1,5,6,7,8].

Fat-Liposoluble Vitamin Deficiencies	Typical and Frequent Clinical Manifestations
Vitamin A	Reduced night vision (night blindness, even blindness), keratomalacia, susceptibility to infection (due to impaired immunity), dry skin, hyperkeratotic papular lesions, and diarrhea (characteristic combination: ophthalmopathy and immune disorders or rickets)
Vitamin D	Mechanical bone pain and fractures, Looser–Milkmann striae, osteomalacia, rickets, muscular weakness, and susceptibility to falls
Vitamin E	Neuropathy and impaired night vision, leading to dementia and ophthalmoplegia
Vitamin K	Spontaneous hemorrhagic syndrome or following minor trauma, osteopenia, and muscle weakness (characteristic combination: hemorrhagic syndrome and osteopenia)

**Table 3 jcm-13-03641-t003:** Treatment regimens for fat-soluble vitamin deficiencies [1,18,19,20,21].

Fat-Soluble Vitamin Deficiency	Curative Treatment
A	100,000 IU/day for 3 days, then 50,000 IU/day for 2 weeks
D	cholecalciferol 3000–10,000 IU/day (average 5000 IU/day) for 1–2 months, or cholecalciferol in a single dose of 60,000 to 150,000 IU, followed by the maintenance dose (2000 IU/day)
E	50–2000 IU/day
K	Treatment is limited to neonates and anti-vitamin K overdoses

IU: international unit.

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
