# Peer review of "Fat-Soluble Vitamins A, D, E, and K: Review of the Literature and Points of Interest for the Clinician"

_jcm, 2024, doi:10.3390/jcm13133641_

Round 1

Reviewer 1 Report

Comments and Suggestions for Authors

Review of the Manuscript "Fat-soluble vitamins A, D, E, and K: review of the literature and points of interest for the clinician"

The objective of the study was to highlight data of interest to the practitioner concerning fat-soluble vitamins, with particular reference to the most recent literature. The manuscript is well-written, with clear and concise language that enhances the accessibility of the content.

Having thoroughly examined the content, I would like to pose a few clarifying questions to better understand certain aspects of the study.

·         GI disorder that impair fat absorption are very different. How could you combine all these into one vitamin supplementation protocol?

·         Could you elaborate on how different liver diseases alter fat metabolism?

·         What strategies do you suggest to avoid hypervitaminosis? Which steps healthcare providers undertake?

·         How to avoid unnecessary testing for vitamin D?

·         You mentioned individualized protocols. What are the main challenges and potential solutions?

Kindly incorporate the responses within the manuscript to augment its overall quality.

Author Response

Thanks for your comments.

Dear reviewer,

Please find my responses.

GI disorder that impair fat absorption are very different. How could you combine all these into one vitamin supplementation protocol?

  • Line 347-353
  • Gastrointestinal disorders can impair fat absorption through a variety of mechanisms, including: reduced surface area for absorption, with damage to intestinal villi (e.g. in celiac disease, Crohn's disease) or intestinal resection (e.g., in short bowel syndrome) [1,4]. They can also affect fat absorption through decreased digestive enzyme production (e.g. in chronic pancreatitis, cystic fibrosis); bile salt disruption with damage to ileum (e.g. in Crohn's disease, short bowel syndrome); and intestinal dysbiosis (disturbance of intestinal flora) [1,4].

Could you elaborate on how different liver diseases alter fat metabolism?

  • Line 360-367
  • Liver diseases disrupt fat metabolism by affecting bile production and secretion, lipoprotein metabolism, beta-oxidation of fatty acids, cholesterol and triglyceride synthesis, and ketone body production [1,4]. These disturbances can lead to fat accumulation in the liver, fat malabsorption and systemic lipid imbalances, influencing the patient's overall health.

    All liver diseases accompanied by chronic cholestasis, as primary biliary cirrhosis and sclerosing cholangitis, may also be accompanied by a deficiency of fat-soluble vitamins [1,4].

What strategies do you suggest to avoid hypervitaminosis? Which steps healthcare providers undertake?

  • Line 525-528
  • To avoid hypervitaminosis, it's crucial to carefully assess vitamin requirements, educate patients, prescribe appropriate doses, and regularly monitor patients' health [19,20]. Healthcare professionals must adopt prudent prescribing practices, ensure clear communication between caregivers, and react promptly to signs of vitamin toxicity.

How to avoid unnecessary testing for vitamin D?

  • Line 440-445
  • In practice, to avoid unnecessary vitamin D testing, it is essential to follow clear guidelines, target testing to at-risk populations, prescribe tests thoughtfully and educate patients about appropriate indications [1,4]. Above all, it is essential to reassess the need for repeat testing in patients who have already been diagnosed and treated for vitamin D deficiency, based on their response to treatment and the absence of persistent symptoms. Routine tests should also be avoided in healthy individuals.

You mentioned individualized protocols. What are the main challenges and potential solutions?

  • Line 464-471
  • Implementing individualized protocols to treat vitamin deficiencies is complex, but essential for optimal patient management [1,18]. These protocols must take into account: the variability of individual needs, the complexity of interactions between nutrients, the need for accurate diagnosis of vitamin deficiency, patient compliance, and more generally, economic considerations and the variability of care systems [5,18]. Against this backdrop, the need for forward-looking clinical studies has never been greater. Continuing education, interdisciplinary collaboration and the use of information and communication technologies are key to overcoming these challenges.

Kind regards

Reviewer 2 Report

Comments and Suggestions for Authors

The topic of this paper is very interesting because vitamin deficiency is still a widespread problem. the structure of the work is well done and comprehensive.

However, I have some observations to make:

1 in the materials and methods we talk about having selected publications from the last 5 years but there are 9 publications out of 20 that were published before 2016 (including some from 2001, 2005, 2007).

In particular, the therapeutic regimen for vitamin D are not updated and this is very misleading. I recommend reading guidelines similar to this Italian one: "Definition, Assessment, and Management of Vitamin D Inadequacy: Suggestions, Recommendations, and Warnings from the Italian Society for Osteoporosis, Mineral Metabolism and Bone Diseases (SIOMMMS). Nutrients. 2022 Oct 6 ;14(19):4148.

A recent review (2024) on vitamin D as an hall mark of aging which refers to the effect on the genome, described in the article, was also published on the effects of vitamin D

2. In the initial description it should be explained how Vitamin D is different from other vitamins, so much so that it is considered a prohormone because it has various active and inactive metabolites and because its activation responds to feedback with blood calcium levels.

3 It should also be explained that the toxic effects described are linked to the use of the active metabolite...i.e. calcitriol and not to the recommended one which is D3. D2 (ergocalciferol) whose supplementation scheme is described, however, is not used in Europe. You should reassure yourself about the toxicity of D3 instead of mentioning it in the abstract

4 in table 2 the word "osteomalacia" should not be in brackets because it is an important consequence of Vit D deficiency.

5 Line 399: Is the author sure that osteopenia can be detected simply with a physical examination of the patient and not with x-rays?

Author Response

Dear Reviewer,

Thanks for your comments.

Please find my responses.

1. "In the materials and methods we talk about having selected publications from the last 5 years but there are 9 publications out of 20 that were published before 2016 (including some from 2001, 2005, 2007)."

  • Line 65-66.
  • A number of reference articles on this topic were also used and cited in the references, even though they date from the last twenty years.

"In particular, the therapeutic regimen for vitamin D are not updated and this is very misleading. I recommend reading guidelines similar to this Italian one: "Definition, Assessment, and Management of Vitamin D Inadequacy: Suggestions, Recommendations, and Warnings from the Italian Society for Osteoporosis, Mineral Metabolism and Bone Diseases (SIOMMMS). Nutrients. 2022 Oct 6 ;14(19):4148."

  • This reference is added and integrated in the discussion.
  • Treatment of vitamin D is corrected in light of this reference.

"A recent review (2024) on vitamin D as an hall mark of aging which refers to the effect on the genome, described in the article, was also published on the effects of vitamin D."

  • This reference is added.
  • Donati S, Palmini G, Aurilia C, Falsetti I, Marini F, Giusti F, Iantomasi T, Brandi ML. Calcifediol: Mechanisms of Action. Nutrients 2023, 15, 4409.

2. "In the initial description it should be explained how Vitamin D is different from other vitamins, so much so that it is considered a prohormone because it has various active and inactive metabolites and because its activation responds to feedback with blood calcium levels."

  • Line 91-93.
  • It should be stressed that vitamin D is different from other fat-soluble vitamins, so much so that it is considered a pro-hormone because it has various active and inactive metabolites, and because its activation responds to feedback from blood calcium levels.

3. "It should also be explained that the toxic effects described are linked to the use of the active metabolite...i.e. calcitriol and not to the recommended one which is D3. D2 (ergocalciferol) whose supplementation scheme is described, however, is not used in Europe. You should reassure yourself about the toxicity of D3 instead of mentioning it in the abstract"

  • Correction are realized.
  • Line 513-516.
  • The above toxic effects described are linked to the use of the active metabolite of vitamin D, namely calcitriol. This is usually not the case with the use of vitamin D3 supplementation, except in cases of other causes of associated hypercalcemia (e.g., use of thiazidic diuretics, use of lithium, Burnett’s syndrome).

  • "For the practitioner, we must not forget the toxicity of excess vitamins, such as hypercalcemia for vitamin D": this sentence is deleted in the abstract

4. "In table 2 the word "osteomalacia" should not be in brackets because it is an important consequence of Vit D deficiency."

  • OK; Done.

5 Line 399: Is the author sure that osteopenia can be detected simply with a physical examination of the patient and not with x-rays?

  • Line 437

    • Corrections are done.
    • "In this context, biological phosphocalcic tests are essential, as well as medical imaging (e.g., bone X-rays or scans) to assess bone density and detect abnormalities associated with vitamin D deficiency [6]."

Round 2

Reviewer 2 Report

Comments and Suggestions for Authors

I am satisfied with the changes made,

only the article that refers to the effects of vitamin D refers instead of Vitamin D to Calcifediol. I was referring to the following review:

Targeting the Hallmarks of Aging with Vitamin D: Starting to Decode the Myth. Nutrients. 2024

Author Response

Dear Reviewer,

Thanks for your suggestion

I have added this reference (Ruggiero, C.; Tafaro, L.; Cianferotti, L.; Tramontana, F.;
Macchione, I.G.; Caffarelli, C.; Virdis, A.; Ferracci, M.; Rinonapoli, G.;
Mecocci, P.; et al. Targeting the Hallmarks of Aging with Vitamin D:
Starting to Decode the Myth. Nutrients 2024, 16, 906).

Kind regards

E. ANDRES
